# Clade diversification dynamics and the biotic and abiotic controls of speciation and extinction rates

Robin Aguilée[1], Fanny Gascuel [2,3], Amaury Lambert [2,4] & Regis Ferriere [3,5,6]

How ecological interactions, genetic processes, and environmental variability jointly shape the evolution of species diversity remains a challenging problem in biology. We developed an individual-based model of clade diversification to predict macroevolutionary dynamics when resource competition, genetic differentiation, and landscape fluctuations interact. Diversification begins with a phase of geographic adaptive radiation. Extinction rates rise sharply at the onset of the next phase. In this phase of niche self-structuring, speciation and extinction processes, albeit driven by biotic mechanisms (competition and hybridization), have essentially constant rates, determined primarily by the abiotic pace of landscape dynamics. The final phase of diversification begins when intense competition prevents dispersing individuals from establishing new populations. Species' ranges shrink, causing negative diversity-dependence of speciation rates. These results show how ecological and microevolutionary processes shape macroevolutionary dynamics and rates; they caution against the notion of ecological limits to diversity, and suggest new directions for the phylogenetic analysis of diversification.

[1] Laboratoire Evolution et Diversité Biologique, UMR 5174, Université Paul Sabatier, CNRS, IRD, 118 route de Narbonne, 31062 Toulouse cedex 9, France. [2] Center for Interdisciplinary Research in Biology (CIRB), Collège de France, 11 place Marcelin Berthelot, 75005 Paris, France. [3] Institut de Biologie de l'École Normale Supérieure (IBENS), Ecole Normale Supérieure, 46 rue d'Ulm, 75005 Paris, France. [4] Laboratoire de Probabilités, Statistique et Modélisation (LPSM), Sorbonne Université, Case courrier 158, 4 Place Jussieu, 75005 Paris, France. [5] Department of Ecology and Evolutionary Biology, University of Arizona, Tucson, AZ 85721, USA. [6] International Center for Interdisciplinary and Global Environmental Studies (iGLOBES) UMI 3157 CNRS, ENS - PSL University, University of Arizona, 845 N Park Avenue, Tucson, AZ 85719, USA. These authors contributed equally: Robin Aguilée, Fanny Gascuel. These authors jointly supervised this work: Amaury Lambert, Regis Ferriere. Correspondence and requests for materials should be addressed to R.F. (email: ferriere@biologie.ens.fr)

Macroevolution is the long-term process that shapes large-scale patterns of species diversity. Since the Modern Synthesis, a debated question has been whether macroevolutionary dynamics are driven primarily by abiotic factors, such as geological changes in the landscape, or by biotic processes, such as ecological interactions within and between species. In the "Red Queen" hypothesis[1] as well as in Darwin's original "principle of divergence"[2,3], ecological interactions are the dominant drivers of macroevolution. In the alternate, so-called "Court Jester" scenario[4], macroevolutionary dynamics are dominated by clade-wide effects of abiotic changes in the physical environment. Although the interplay between biotic and abiotic factors has long been recognized as fundamental in macroevolution, progress toward understanding this interaction has been slow.

Decomposing the macroevolutionary dynamics of a clade into constituent parts—speciation and extinction—is a useful step toward a mechanistic understanding of the drivers of clade diversification. As statistical models become increasingly available to analyze patterns of variation in speciation and extinction rates from reconstructed phylogenies[5,6], these phylogeny-based models are used to test for time-dependence[7,8], environment-dependence, diversity-dependence[9,10], or trait-dependence[11–13] of macroevolutionary rates. However, the results of such analyses are often limited by the assumption that a single process, such as environmental or diversity dependence, drives clade diversification, and leaves unexplained why these dependencies might occur[6,14]. Further progress has been impeded by the lack of mechanistic theory linking macroevolutionary dynamics and the potential biotic and abiotic factors of speciation and extinction rates[6,15–18].

Mechanistic models of clade diversification that integrate biotic and abiotic factors have begun to emerge[19–23]. These models incorporate genetic mechanisms of reproductive isolation in evolving populations of interacting individuals, and thus allow to relate the process of speciation to ecological characteristics of populations and physical (e.g., geographical) properties of the environment. Here, we further develop the model introduced in Aguilée et al.[21] and Gascuel et al.[23] to elucidate whether and how macroevolutionary diversification and the underlying rates of speciation and extinction are controlled by the abiotic factors of landscape geographical structure and dynamics, and the biotic factors of resource competition and genetic differentiation.

The model describes the evolutionary diversification of a clade initiated from a single ancestral species living on a two-dimensional landscape. The landscape is fragmented in habitat patches or sites. The baseline version of our model assumes a resource gradient in each direction of the landscape; we also test the robustness of the resource-gradient scenario by randomizing the resource distribution across sites (Supplementary Fig. 1). Individuals are characterized by two phenotypic traits measuring their capacity to use each resource. Evolution is driven by (i) heritable variation in the traits, (ii) selection shaped by local competition (within geographically connected patches), (iii) migration between geographically connected patches, and (iv) genetic drift at loci where mutation can create neutral, genetically incompatible alleles.

Organisms reproduce sexually and have differentiated sexes, and we use the biological species concept[24] (Supplementary Fig. 2). The model accounts for two fundamental genetic mechanisms of speciation: prezygotic reproductive isolation can evolve as selection against maladapted hybrids favors assortative mating; postzygotic isolation can evolve as genetic incompatibilities accumulate between populations[24–27]. To control for the influence of dispersal on gene flow, we keep the individual dispersal capacity constant across time, space, and species. Thus, time and space variation in gene flow between populations is solely determined by variation in reproductive isolation (due to individual genotypes), and external (geographical) barriers to dispersal—not by individual variation in dispersal capacity.

An important parameter in the model is competition width, defined as the range of resources that a species can use compared to the range of resources present locally within a site. By assuming constant competition width, we control for the effect that evolving specialization may have on diversification. In other words, keeping competition width constant ensures that the evolution of specialization does not contribute to the diversification of the clade.

With the Modern Synthesis, speciation research has emphasized the role of abiotic factors for the major effect they can have as physical barriers to gene flow or "openers" of ecological opportunities[24,26]. The model captures the essence of both, with the incorporation of fluctuating geographical barriers[19,21], and local catastrophes that can reopen environmental niches[23]. Geographical barriers randomly arise and disappear across the landscape, thus causing the recurrent isolation or connection of geographical sites where local communities (interacting populations) live. Local catastrophes are major disturbances that strike individual sites at random in time and space, and wipe out all populations at once in the impacted sites. The expected times that any two neighboring sites remain isolated from one another or connected set the temporal scale of the landscape dynamics. The spatial scale of the landscape is set by the individual dispersal capacity, in the sense that individuals of all species present in adjacent sites can disperse freely between these sites when the latter are connected; when a barrier arises between the two sites, dispersal is interrupted. Within connected sites, the multilocus genetic basis of the evolving traits makes sympatric speciation very unlikely[28,29].

Across a broad range of parameters, the model shows diversification unfolding in three distinct stages. Clade diversification begins with an adaptive radiation, during which the speciation rate is controlled primarily by geographic isolation and the buildup of genetic incompatibilities between isolated populations. In the second phase of diversification, the processes of speciation and extinction are enacted by biotic mechanisms (competition and hybridization); the speciation rate is constant, determined chiefly by the pace of landscape dynamics. In the final phase, the model predicts evolutionary slowdown to a stationary state at which species diversity is controlled by resource competition. Past the initial adaptive radiation, the extinction rate is approximately constant and determined primarily by the pace of landscape dynamics and the number of genetic incompatibility loci.

To explain these results, we monitor several key ecological and genetic components of speciation and extinction throughout the course of macroevolutionary diversification: species abundance, geographic range, competition intensity, local adaptation, phenotypic divergence, assortative mating, and hybridization risk. By analyzing how these key variables vary and covary with three biotic factors (the number of genetic incompatibility loci, competition width, and resource abundance) and three abiotic factors (pace of landscape dynamics, geographical isolation time, and rate of local catastrophe), we establish causal relations between these factors and the emerging macroevolutionary rates and dynamics. Our results explain how microevolutionary processes scale up and shape the macroevolutionary dynamics of species diversity. They call for phylogenetic analyses that distinguish between different temporal stages of diversity-dependence and diversity-independence and enable testing predictions regarding the ecological, genetic, and environmental controls of macroevolutionary rates in each stage.

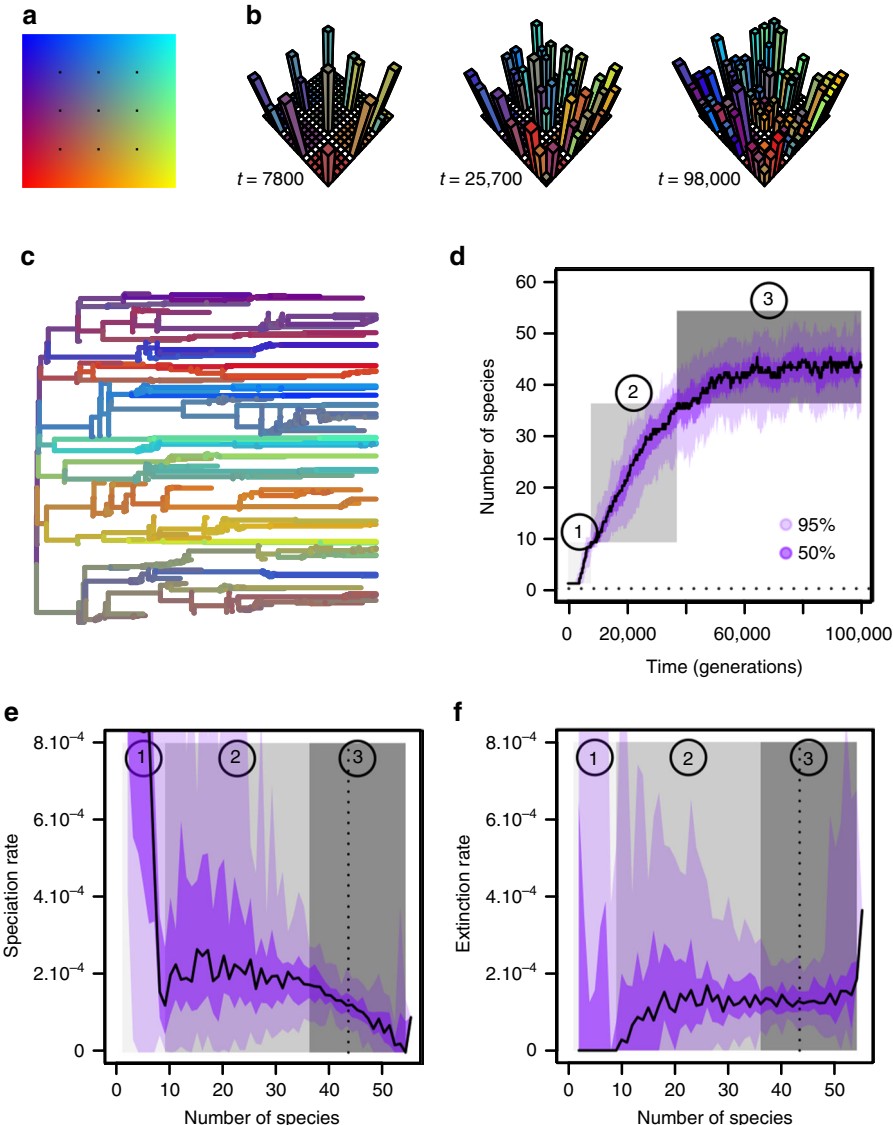

**Fig. 1** Macroevolutionary dynamics in silico. **a**, **b** Example of diversification in space and time. Color-coded phenotypes vary continuously across a two-dimensional phenotypic space (**a**); black dots indicate the phenotypic optimum in each of the nine geographic sites. Panel **b** shows the populations (vertical bars, height measures abundance, and color indicates phenotype) of all species at three different times ($t$), across the nine geographical sites into which the landscape (horizontal plane) is subdivided. The color of each site indicates the corresponding phenotypic optimum. **c** Phylogenetic tree of extant and extinct species, with colors showing the average phenotype along each branch. **d** Species diversity through time, measured in the number of generations from the common ancestor's introduction in the landscape. **e** Speciation rate as a function of diversity. **f** Extinction rate as a function of diversity. In **d**–**f** black lines give median values over 50 simulation replicates, purple areas give 95% (dark purple) and 50% (light purple) confidence intervals, and the three stages of diversification (1–3) are highlighted by gray shadings. Parameter values as in Supplementary Table 1

## Results

**Early geographic adaptive radiation**. Exposing a single ancestral species to new environmental opportunities ignites diversification with an adaptive radiation (Fig. 1), associated with high speciation rates (Fig. 1e, stage 1) and rare extinction (Fig. 1f, stage 1). Rapid character divergence (as exemplified in Fig. 1a–c, Supplementary Fig. 10a–b) is driven mainly by local adaptation within sites that are geographically isolated, and the evolution of postzygotic reproductive isolation. This is supported by four lines of evidence. First, the radiation speed is insensitive to competition width (Fig. 2a and Supplementary Fig. 4b), and the overall intensity of interspecific competition remains low regardless of competition width (Fig. 3a and Supplementary Fig. 6, Interspecific competition vs. Competition width). Second, local adaptation is limited by gene flow when the isolation time is

short; we therefore expect the radiation to be impeded by decreasing the isolation time, which is the case (Fig. 2a, Supplementary Fig. 4q, Supplementary Fig. 6, Maladaptation vs. Time in isolation). Third, we predict a faster radiation if directional selection toward local environmental optima is stronger, which occurs when more abundant resources result in larger local population size; we find that increasing resource abundance does accelerate the radiation, as expected (Fig. 2a, Supplementary Fig. 4c, Supplementary Fig. 6, Abundance per site vs. Resource abundance). Finally, assortative mating is low (Fig. 3e) and when the genetic distance (the number of incompatible genetic loci) needed to induce postzygotic reproductive isolation between populations is smaller, we observe earlier diversification (Fig. 2a, Supplementary Fig. 4a, Supplementary Fig. 7b, d). We thus conclude that if genetic incompatibility among allopatric

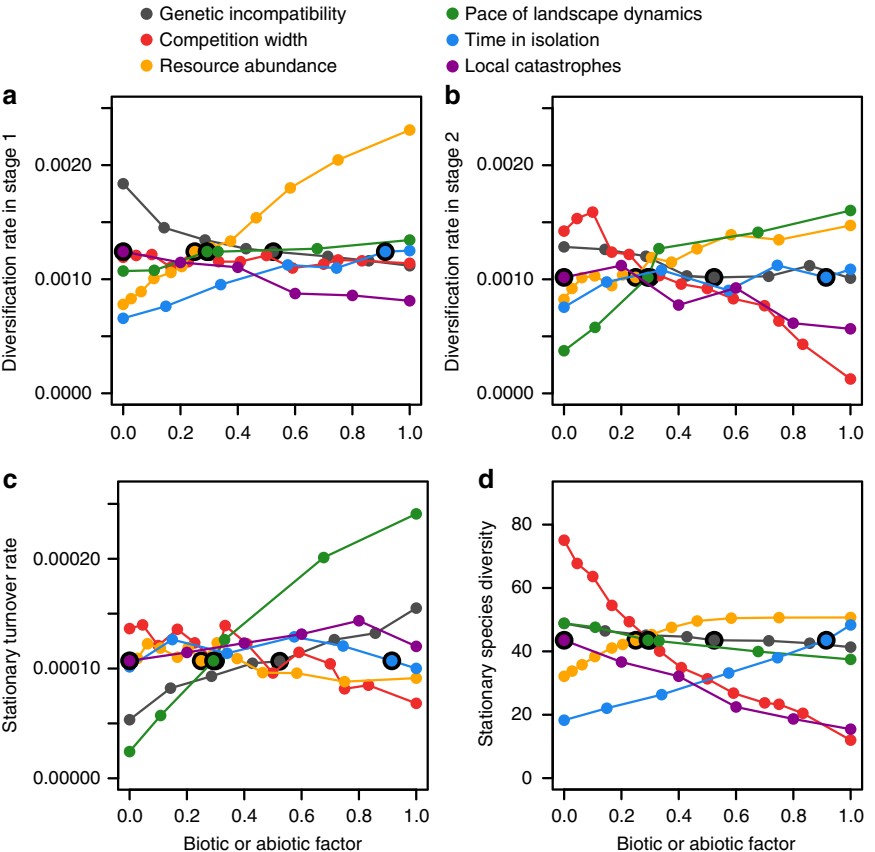

**Fig. 2** Effects of different biotic and abiotic factors on macroevolutionary rates and species diversity. **a** Diversification rate in stage 1. **b** Diversification rate in stage 2. **c** Stationary turnover rate. **d** Stationary species diversity. Values are medians over 50 simulation replicates. See text and Methods for factor definitions. The range of parameter values used for each factor was rescaled between 0 and 1, with 0 (respectively 1) corresponding to the smallest (respectively highest) parameter value used for that factor (see Methods). Black open circles correspond to default parameter values (Supplementary Table 1). Other parameter values as in Supplementary Table 1

populations can build up quickly relative to the sites' isolation time, ecological speciation is of minor importance in the initial radiation.

**Niche self-structuring drives diversification phase 2.** Diversification does not stop with one locally adapted species filling each environmental niche. As the sequence of events of isolation/connection of sites repeats itself through time and between different pairs of sites, the clade enters a second phase of diversification (stage 2 in Fig. 1d–f, Fig. 2b, Supplementary Fig. 10c–e). This occurs through a process of niche self-structuring[30] whereby species evolve away from local environmental optima (Fig. 3g, Supplementary Fig. 6, Maladaptation), while their geographic range shrinks and population size decreases (Fig. 3i, k and Supplementary Fig. 6, Geographic range size, Abundance per site). Overall, the intensity of interspecific competition increases (Fig. 3a and Supplementary Fig. 6, Interspecific competition). Diversification by niche self-structuring leads to the coexistence of many more species than environmental optima (Figs. 1d and 2d).

While the rate of the initial adaptive radiation was essentially determined by isolation time (a measure of geographic isolation), resource abundance (which sets the strength of selection toward local optima), and genetic incompatibility loci (shaping the evolution of postzygotic isolation), the speed of diversification in the niche self-structuring phase is most sensitive to competition width and landscape dynamics (Fig. 2b). Narrow competition width (i.e., high resource specialization) or a fast enough

landscape dynamics can make diversification even faster in the niche self-structuring phase compared to the earlier adaptive radiation (Fig. 2a, b); in contrast, large competition width (i.e., low resource specialization) or slow landscape dynamics make it exceedingly slow (Fig. 2b).

To identify the drivers of speciation and extinction during the niche self-structuring phase of diversification, we monitored (Fig. 3) all the process variables and compared them among species that are about to speciate, species about to go extinct, and the remaining species (which we call "static"). We also evaluated the effect of all biotic and abiotic factors on speciation and extinction rates (Supplementary Fig. 5) and on the above variables (Supplementary Figs. 6 and 8).

**Controls of speciation rate during niche self-structuring.** Speciation proceeds at a constant rate during diversification by niche self-structuring (stage 2 in Fig. 1e). Speciation events are associated with a large enough geographic range extending over multiple sites (Fig. 3j), which correlates with a relatively small local population size (Fig. 3l) and a relatively high degree of environmental maladaptation (Fig. 3h). The rate of speciation is influenced predominantly by the landscape dynamics (Supplementary Fig. 5k), and secondarily by competition width (Supplementary Fig. 5e). More frequent isolation promotes local adaptation, which favors between-site divergence. With narrower competition, a species can have a larger geographic range (i.e., occurs on more sites, Supplementary Fig. 6, Geographic range vs. Competition width) by evolving the use of more marginal

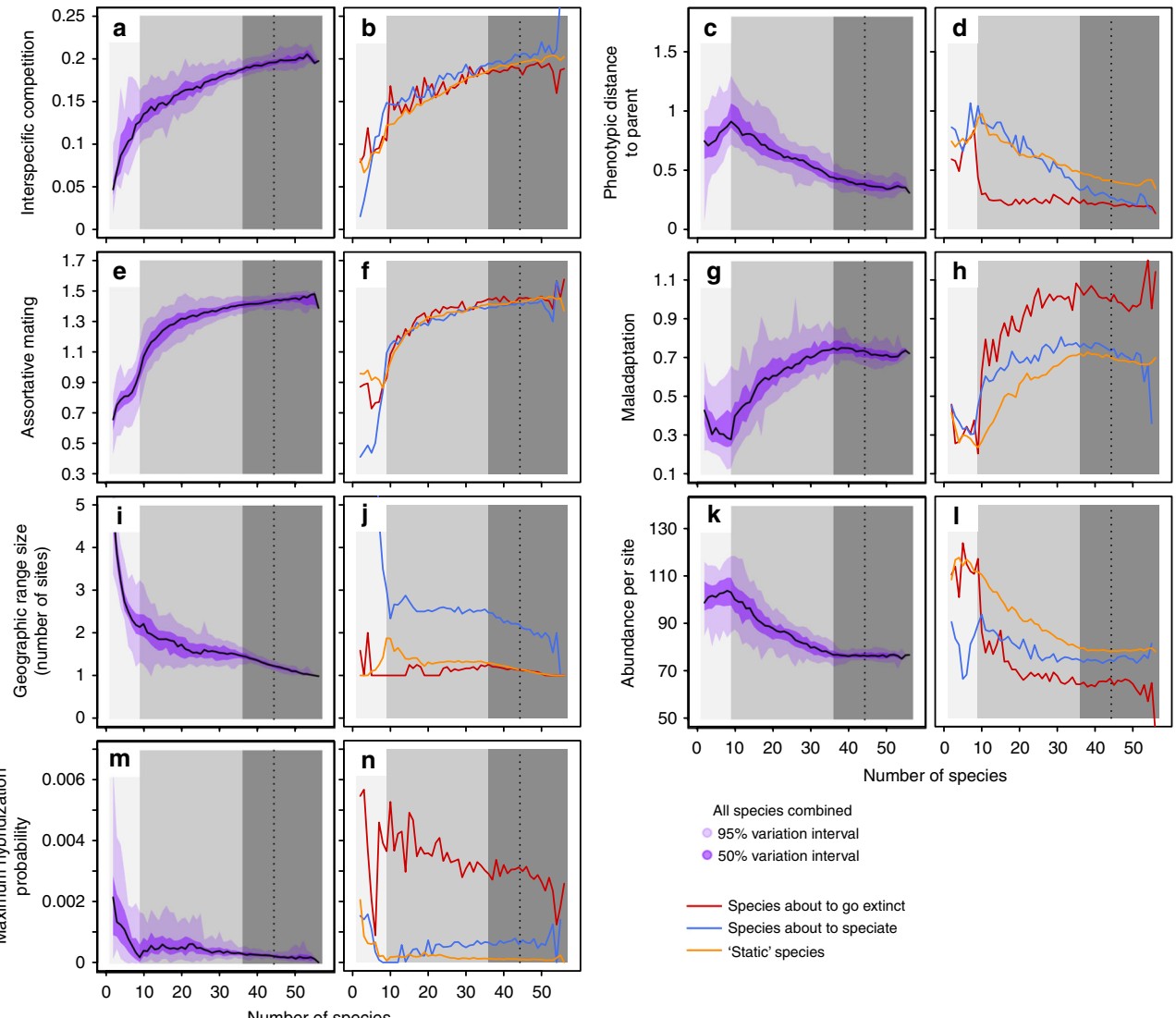

**Fig. 3** Diversity dependence of key ecological and genetic variables. Simulations were replicated 50 times with default parameter values (Supplementary Table 1). For each variable (see Methods, section "Numerical simulations" for details about the computation of each variable), the left column (**a**, **c**, **e**, **g**, **i**, **k**, **m**) shows median values (black lines), 95% and 50% confidence intervals (purple areas), computed across all species (as in Fig. 1d–f); the right column (**b**, **d**, **f**, **h**, **l**, **n**) shows median values for species about to speciate (within the next 200 generations; blue), about to go extinct (within the next 200 generations; red), and for "static" species (neither speciation nor extinction within the next 2000 generations; orange). The three stages of diversification (1–3) are highlighted by gray shadings, as in Fig. 1. A sensitivity analysis of these results is reported in Supplementary Figs. 6 and 8

resources in each site (a species may use resources that are locally rare but present in up to four sites, in which the species may thus occur); this facilitates speciation (Supplementary Fig. 5e, Supplementary Fig. 8, Geographic range size vs. Competition width) while increasing the overall degree of environmental maladaptation (Supplementary Fig. 6 and 8, Maladaptation vs. Competition width).

Resource abundance has surprisingly little effect on the speciation rate (Supplementary Fig. 5h) in spite of being a decisive influence of local adaptation. Increasing resource abundance makes within-site selection toward the local optimum more effective and thus favors character divergence between sites. However, as local population sizes increase, interspecific competition intensifies, hampering character convergence expected within sites from local adaptation; this tends to reduce the between-site divergence of geographic pairs of species. These antagonistic effects appear to balance one another out and leave the speciation rate essentially insensitive to resource abundance.

The average degree of homogamy increases sharply early on but soon approaches saturation (Fig. 3e). Also, on average, species about to split show neither an excess nor a deficit of homogamy, compared to static species (Fig. 3f). In comparison, the fraction of species pairs in which reproductive isolation evolves by genetic incompatibility is consistently small past the initial radiation (Supplementary Fig. 7i–j). As a consequence, the macroevolutionary rate of speciation is decoupled from the rate of reproductive isolation evolving by homogamy or the buildup of genetic incompatibility[31,32].

**Controls of extinction rate**. After the initial radiation drives species diversity above the number of environmental niches, the extinction rate rises sharply and stabilizes. From then on, most species going extinct are young (Supplementary Fig. 9c). Young species may have a lower population size, and low population size is indeed a factor of extinction (Fig. 3l). However, the abundance

of species destined to extinction is typically no less than half the abundance of static species (Fig. 3l), which shows that extinction is not merely induced by populations stochastically drifting to a small size. Young species also tend to be little differentiated from their mother species, and insufficient character divergence is indeed key to extinction (Fig. 3d). The relatively low abundance of species about to go extinct is a side effect of their tendency to occur in a single site (Fig. 3j) to which they are poorly adapted (Fig. 3h).

The stationary extinction rate is determined primarily by the pace of landscape dynamics, and secondarily by the number of genetic incompatibility loci inducing postzygotic reproductive isolation (Fig. 2c, Supplementary Fig. 4, Stationary turnover, Supplementary Fig. 5, Extinction rate). Site isolation exposes new species that are poorly differentiated from their mother species to be excluded competitively because they are too maladapted in their new habitat (Fig. 3d, h, Supplementary Fig. 10g–h). Following the connection of two sites, secondary contact with the mother species can also trigger competitive exclusion (Supplementary Fig. 10f–g). Moreover, limited divergence may cause reinforcement to fail when sites are reconnected; nascent species may then be lost by hybridization with their mother species (Fig. 3n, Supplementary Fig. 10f–g). The influence of the number of genetic incompatibility loci inducing reproductive isolation on the extinction rate (Fig. 2c, Supplementary Fig. 4c, Supplementary Fig. 5c, and Supplementary Fig. 7f, h) is a manifestation of the importance of hybridization on extinction: with fewer genetic incompatibility loci, postzygotic reproductive isolation evolves faster, which reduces the risk of hybridization and thus lowers the extinction rate (Supplementary Figs. 6 and 8, Maximum hybridization probability vs. Genetic incompatibility).

Although competition width and resource abundance are key factors of divergence, they only have a small influence on the extinction rate (Fig. 2c, Supplementary Fig. 4g, k, Supplementary Fig. 5f, i). More divergence generally reduces the risk of extinction from competitive exclusion or hybridization at secondary contact, but enhances the level of maladaptation in subsequent bouts of geographic isolation; these two opposing effects appear to balance out, leaving the extinction rate unaltered by these ecological factors.

**Speciation slowdown and extinction blowup.** When the phase of diversification by niche self-structuring ends, all species are distributed in one or two sites only. During the third phase of diversification (stage 3 in Fig. 1d–f and Supplementary Fig. 3), it is essentially two-site species that contribute to speciation (Fig. 3j). Each speciation event then leaves both mother and daughter species confined to single sites. When sites become reconnected, extreme maladaptation (Fig. 3g) and intense interspecific competition (Fig. 3a) prevent recolonization and the establishment of new populations. Rare speciation events occurring within a single site (Fig. 3j) may result from the ephemeral phenotypic differentiation of subpopulations of an abundant species peaking at or very close to a local environmental optimum (Fig. 3h, l).

Negative diversity-dependence of the speciation rate (Fig. 1e, Supplementary Fig. 5, Speciation rate) is thus caused by a "geographic ratchet", whereby species range shrinks irreversibly as diversity increases. This pattern is enhanced by conditions of low resources (which reduces the chance of establishment of new populations; Supplementary Fig. 5h), large competition width (which constrains divergence; Supplementary Fig. 5e), or fast landscape dynamics (which sets the pace at which new populations are seeded; Supplementary Fig. 5k). Because homogamy is strong (Fig. 3e, Supplementary Fig. 6, Assortative mating), the role of genetic incompatibility in this phase is

essential to maintain reproductive isolation between species with ancient divergence (Supplementary Fig. 7i–j, Supplementary Fig. 9a); varying the number of incompatibility loci inducing reproductive isolation has thus little impact on the diversity-dependent speciation rate (Supplementary Fig. 5b).

When diversity reaches a critical number, the speciation rate nears zero and the extinction rate blows up (Fig. 1e, f, Supplementary Fig. 3, Supplementary Fig. 5, Speciation rate and Extinction rate). This critical diversity arises as a further consequence of the geographic ratchet—it is the diversity at which *every* species is restricted to a single site (Fig. 3i). At this point, each connection event will decrease the density of newly co-occurring species because of previously absent, interspecific competitive interactions. As sites split again, the newly seeded subpopulations are unlikely to form new species or even persist; meanwhile, the "mother" populations that have been significantly reduced during the previous sympatry phase now face a high chance of extinction. Faster isolation–connection cycles are thus expected to exacerbate the extinction blowup and this indeed is the case (Supplementary Fig. 5l). Additional extinction may also follow from the rare speciation events happening at critical diversity. Speciation then only concerns species located phenotypically at or very close to their local environmental optimum (Fig. 3h). Some of the speciation events are driven by hybridization (Fig. 3n) which entails the loss of one of the parental species (or both); if the mother species persist, daughter species are unable to diverge significantly and face a high risk of quick extinction due to the strong interspecific competition that prevails near the optimum. Moreover, intense competition experienced by incipient species (Fig. 3b) is likely to affect both mother and daughter species immediately after speciation.

**Controls of long-term diversity.** In spite of the random nature of landscape dynamics and catastrophes, for a given set of parameters, the clade's diversity reaches a predictable, stationary number of species, or long-term diversity (Figs. 1d and 4 and Supplementary Fig. 5, Number of species). As the clade diversifies, the size of the metacommunity (total number of individuals of all species present in the landscape) increases. Clade diversity and metacommunity size fluctuate within a narrow range around their stationary value. The species rank-abundance curve of the metacommunity stabilizes (Fig. 4g–l), while species composition changes at a stationary turnover rate (Fig. 2c, Supplementary Fig. 4, Stationary turnover rate). Long-term diversity can exceed the number of environmental optima by far (grand average long-term diversity across all simulations ca. 45 vs. 9 environmental optima) but may fall well below critical diversity (grand average across all simulations ca. 55). In the stationary state, the overall intensity of interspecific competition, level of environmental maladaptation, and homogamy are close to a maximum (Fig. 3a, e, g), while phenotypic divergence, geographic range size (number of sites), and species abundance are close to a minimum (Fig. 3c, i, k).

In our simulations, both stationary and critical diversity are correlated and predicted primarily by competition width and secondarily by resource abundance, time in isolation, and catastrophe rate (Fig. 2d, Supplementary Fig. 4, Stationary species diversity, and Supplementary Fig. 5, Number of species). This is in contrast to the stationary turnover rate, which varies primarily with the pace of landscape dynamics (especially the rate of site disconnection, given the small influence of isolation time) and secondarily with the number of genetic incompatibility loci inducing reproductive isolation (see "Controls of extinction rate"). Increasing competition width or shortening the isolation time when every species is restricted to a single site (i.e., when the

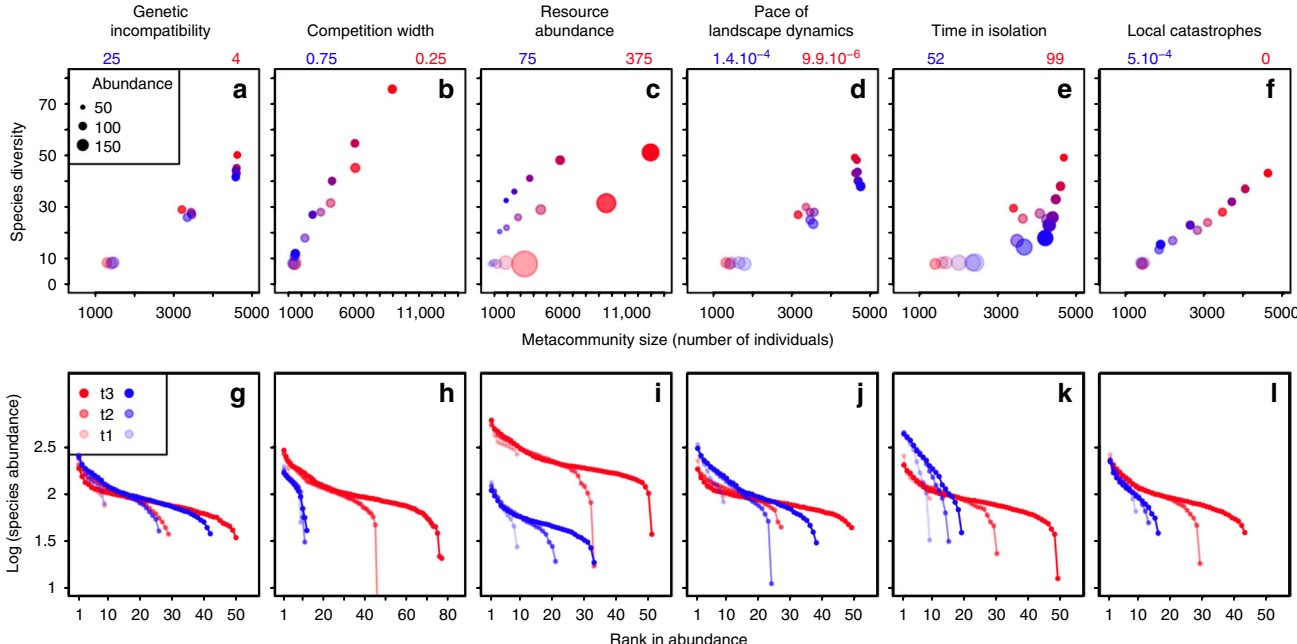

**Fig. 4** Effect of biotic and abiotic factors on the evolutionary emergence of macroecological patterns. **a–f** Relationship between species diversity and metacommunity size. **g–l** Species rank-abundance distributions. For each biotic and abiotic factor, the median values of the macroecological variables of interest (over 50 simulation replicates) are shown at different stages of diversification: at the end of stage 1 (light color), mid-way through stage 2 (medium), and at a stationary state (dark). Factor values are scaled from blue (leading to the lowest stationary species diversity) to red (leading to the highest stationary species diversity), with the corresponding values indicated above each column. Other parameter values as in Supplementary Table 1. In **a–f** the size of dots is proportional to the median value of species abundances. See text and Methods for the definition of each factor

clade is at critical diversity) exposes each species to more competition when the species' site is reconnected to a neighboring site, due to either greater niche overlap or more time spent in competition (Supplementary Fig. 6, Interspecific competition vs. Competition width, Phenotypic distance to the parent, and Maladaptation vs. Time in isolation); decreasing resource abundance makes populations more prone to extinction (Supplementary Fig. 6, Abundance per site vs. Resource abundance). The critical diversity drops consequently (Supplementary Fig. 5d, g, m). Increasing the rate of local catastrophe mechanically lowers the clade's diversity at which the spatial occupancy of all species is down to a single site (Supplementary Fig. 6, Geographic range size vs. Local catastrophe), i.e., the critical diversity. On the contrary, the pace of landscape dynamics and the number of incompatibility loci do not affect the critical diversity and thus the stationary diversity (Supplementary Fig. 5a, j).

The integration of competition with genetic and abiotic factors of speciation and extinction thus yields the following predictions on long-term macroecological patterns: (1) A quasi-linear positive correlation between long-term diversity and metacommunity size, irrespective of clade age, arises in response to variation in competition width (Fig. 4b); a similar correlation can also be driven abiotically by the catastrophe rate, but is then much shallower (Fig. 4f). (2) Variation in isolation time can cause variation in long-term diversity without substantial change in metacommunity size (Fig. 4e). (3) Variation in resource abundance can cause variation in metacommunity size without substantial change in long-term diversity (Fig. 4c). (4) Variation in either the pace of landscape dynamics or the number of genetic incompatibility loci inducing reproductive isolation impacts species turnover rates (Fig. 2c) but leaves both long-term diversity and metacommunity size unaltered (Fig. 4a, d). Competition width, resource abundance, isolation time, and catastrophe rate may all contribute to explaining variation in long-term diversity

among clades, through their effect on critical diversity; the concurrent change in metacommunity size, however, differs between factors and thus provides a signature of the causal mechanism.

## Discussion

Macroevolutionary models that explicitly take into account ecological, genetic, and environmental processes are needed to advance our mechanistic understanding of variation in macroevolutionary rates of speciation and extinction. Such understanding is critical to explain the emergence and structure of species diversity patterns, and investigate how species diversity may respond to current and future environmental changes and concomitant community composition alterations[33]. Our analysis shows how biotic and abiotic factors interact and influence speciation and extinction rates throughout the process of clade diversification. For the range of parameters used in our simulations, clade diversification occurs in three stages. In the initial adaptive radiation, speciation is essentially geographic, driven by landscape fragmentation, local adaptation, and genetic divergence among isolated sites. Most of the diversification occurs in the second stage, which requires the interaction of competition-mediated character divergence and fluctuations in landscape geographic structure (Supplementary Fig. 10). As ecological characters evolve, niche self-structuring[30] occurs, driving species diversity well above the number of environmental optima. During this phase, the pace of landscape dynamics emerges as the top control of both speciation and extinction rates; the number of genetic incompatibility loci inducing postzygotic reproductive isolation is the next significant control of the extinction rate, which points at hybridization as a significant cause of extinction. The geographic range of every species tends to shrink as the clade diversifies; when most species are restricted to one or two sites (the smallest geographic unit of a habitat in our model), the third

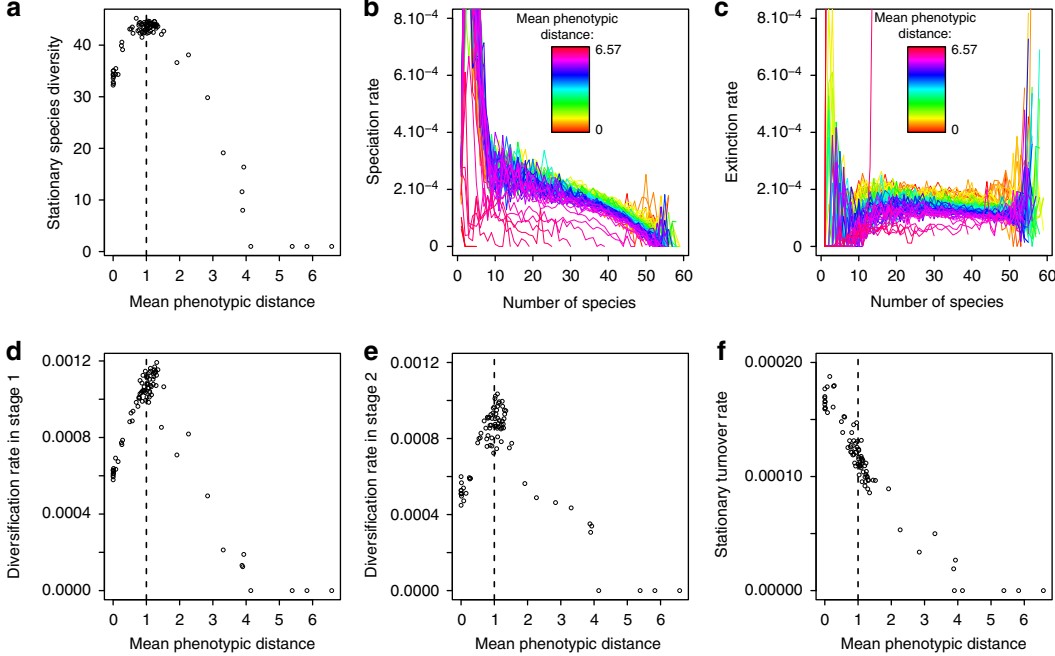

**Fig. 5** Effect of sites distribution and distance between ecological optima on diversification dynamics. We generated 84 landscapes for which the environmental optima of the nine sites are randomly chosen in uniform distributions of various widths. The phenotypic distance between the optima of adjacent sites (and thus the degree of resource distributions overlap of adjacent sites) is different for each pair of sites, and there is no correlation between geographical proximity and ecological similarity (Supplementary Fig. 1). **a** Stationary species diversity as a function of the mean phenotypic distance between ecological optima of adjacent sites. **b** Speciation rate as a function of diversity. **c** Extinction rate as a function of diversity. **d–f** Diversification rate in stage 1 (**d**), in stage 2 (**e**), and stationary turnover rate (**f**) as a function of mean phenotypic distance between environmental optima of adjacent sites. The vertical dashed line indicates the phenotypic distance between environmental optima of adjacent sites in the baseline version of the model. The results are based on 50 simulation replicates for each landscape. Parameter values as in Supplementary Table 1

stage of diversification begins. As diversity keeps rising, competition further intensifies, which makes it increasingly unlikely that one-site species will expand their geographic range. This results in the negative diversity dependence of the speciation rate and the stabilization of the species number below a critical diversity threshold at which all species are endemic to a single site, the speciation rate approaches zero, and the extinction rate blows up.

We expect the following assumptions to be critical for the general pattern of diversification predicted by our model. First, the landscape should consist of several sites and several environmental optima to allow for allopatric divergence during the initial phase of the radiation. Second, the coexistence of several species in sympatry requires that resource distributions are sufficiently wide for the ecological space to support more than one species per habitat type, and that selection is locally disruptive for maintaining or reinforcing ecological divergence in sympatry. Third, resource distributions should overlap between sites to allow the propagation of local diversity when sites are connected during the second phase of diversification[34,35]. Very small and very broad overlaps oppose diversification (Fig. 5a). In particular, a very broad overlap limits the effectiveness of the diversification process because species in connected sites may not be differentiated enough to allow stable coexistence (Fig. 5a).

Within the range of resource distribution overlap allowing diversification, the general predictions of our model hold even when geographical location and ecological similarity of sites are uncorrelated (Fig. 5b, c). In landscapes where the overlap of resource distributions is smaller than in the gradient scenario, diversification rates in stages 1 and 2 are higher, because allopatric divergence then proceeds faster (Fig. 5d, e). A smaller overlap of resource distributions also results in lower stationary turnover. This is because speciation is less likely, due to failed

colonization of adjacent sites; and extinction is less frequent because species at secondary contact are more differentiated (Fig. 5f).

Constant competition width across the diversifying clade is another important assumption. Earlier models by Ackermann and Doebeli[36] and Ito and Shimada[37] addressed the question of how niche width may coevolve with niche position. Their results support a general pattern of a narrower niche evolving across a range of phenotypes that diversify in their niche position. In our model, this would manifest as a gradual reduction of niche width (barring the episodic evolution of more generalist transient species[37]). Our results suggest that decreasing niche width would accelerate speciation in the second phase of diversification, with little effect on extinction rates, leading eventually to higher long-term diversity. We thus expect the main alteration of our results in response to niche width evolution to be the emergence of positive diversity dependence of speciation in the second phase of diversification (as more specialization evolves), and more shallow negative diversity dependence of speciation in the third phase (as predicted for small competition width). Given the relatively small effect of competition width on speciation that our model shows in diversification phase 2, we expect that abiotic factors (the pace of landscape dynamics) would remain the dominant factor of speciation rates (and extinction rates) past the initial radiation. Confirming and refining these expectations warrants future work. The other assumptions of our model may either promote (e.g., assortative mating based on ecological traits under disruptive selection) or hamper (e.g., rare sympatric speciation) diversification, but none of them is expected to be critical to the pattern of diversification described here[21,23].

Our investigation of biotic vs. abiotic controls of macroevolution sheds new light on the scope of Darwin's "principle of

divergence", which lays at the core of Darwin's own approach to macroevolution[3]. The "principle" assumes that new species begin as emerging varieties; the ones that survive the "struggle for existence" are populations that evolve more distinct characteristics and resource use, to the point where they may cause their ancestor's and some of their close relatives' extinction. In this view, reproductive isolation is a by-product rather than a driver of divergence, and the physical environment is of minor importance. The Modern Synthesis rejected Darwin's principle, minimized the role of competition in species origination and loss, and emphasized the interaction of abiotic factors and genetic mechanisms of reproductive isolation in driving the macroevolutionary dynamics of diversification[24,25,38]. This model shows that, in line with Darwin's principle, biotic interactions are key, but their outcome is determined by an interaction with the abiotic "theater" of diversification—the dynamic geographic landscape. The spatial and temporal scales of landscape dynamics shape both macroevolutionary rates of speciation and extinction, but we emphasize that the landscape's relevant spatial scale itself is relative to the organisms' dispersal capacity. As envisioned by Darwin, the frequent extinction of incipient species insufficiently diverged from their parental species plays a key role in diversification (in stages 2 and 3). This general pattern of early extinctions fits the concept of ephemeral speciation[20,39–41], which predicts the rapid extinction of most newly diverging species. Our model backs up the contention that ephemeral speciation may explain observed mismatches between high microevolutionary rates of reproductive isolation and low macroevolutionary rates of speciation[42–44].

In their analysis of a fossil record of Cenozoic planktonic foraminifera, Ezard et al.[45] found macroevolutionary dynamics to depend on the interaction between species' ecology and the abiotic environment (changing climate). They found speciation rates to be more strongly shaped by diversity dependence than by climate change, whereas the reverse was true for extinction rates. Our results lead to the similar conclusion that most of the clade diversification (in phase 2) is driven by the interplay of competition-driven divergence and landscape dynamics. However, variation in both speciation and extinction rates is dominated by the abiotic effect of landscape dynamics, and the negative diversity dependence of speciation should not be reduced to a simple signature of biotic interactions (see below).

Negative diversity-dependence of speciation is conventionally explained by ecological niche filling, and this had led to the controversial notion that diversification might be constrained by "ecological limits". An alternative to ecological niche filling is failure of geographic speciation[15,35]. Our results support a combination, with the former (ecological niche filling) causing the latter (failure of geographic speciation) by preventing the establishment and divergence of new populations. The emergence and shape of speciation negative diversity dependence thus depends on the interplay between the characteristics of competition (niche width) and the spatial scale of the landscape, which itself is determined by the minimal unit of spatial isolation relative to the organism's characteristic dispersal distance. The diversity threshold above which speciation rate becomes negatively diversity-dependent may become arbitrarily high if the size of the unit of spatial isolation is reduced. This might occur in systems where evolution would drive reduced dispersal in the course of diversification—a scenario not allowed in our model. We expect more complexity in the landscape structure (and in the ecological interactions) and the evolution of dispersal to alter the limit that competition seems to impose to diversification in this simple model.

Yet even in its current form, the model yields results that contrast with the "ecological limits" hypothesis. First, a positive correlation between long-term diversity and metacommunity size is not necessarily due to variation in competition––it can also arise abiotically, from variation in the occurrence rate of local catastrophes. Second, variation in isolation time can cause greater diversity while leaving metacommunity size virtually constant. Third, increasing resource abundance does not necessarily cause an increase in long-term diversity––it may result in higher species abundance (hence a larger metacommunity size). Fourth, the "emptying" of environmental niches by local catastrophes does not result in greater long-term diversity, because the positive effect on speciation is offset by the extinction of diverging populations before speciation is complete.

The model makes four testable predictions about macroevolutionary rates and dynamics. (1) Diversification unfolds in three stages, with low extinction rates in the first (geographic adaptive radiation), constant speciation and extinction rates in the second (niche self-structuring), and negatively diversity-dependent speciation rates in the third. (2) Slow diversification and low diversity is expected in landscapes with short isolation time, in which the initial adaptive radiation is more likely to fail. (3) Ecologically similar clades are expected to diversify and turn over faster in landscapes that fragment frequently, and to harbor higher long-term species diversity if fragmentation lasts longer on average. (4) Clades diversifying in similar landscapes are expected to show faster diversification and higher long-term diversity in response to more resources or less competition, and lower turnover only if genetic incompatibility builds up more easily. These predictions could be tested by comparing patterns of diversification of ecologically similar colonizers entering different landscapes, and of ecologically diverse colonizers entering similar landscapes. Such comparisons might be possible using extensive datasets from island archipelagos, such as the Galapagos Islands or the Hawaiian archipelago. Homes to spectacular evolutionary radiations, island archipelagos have diverse histories of landscape structure and dynamics, that make them amenable to comparing ecologically diverse clades within each landscape, and ecologically similar clades across different landscapes[46].

In this perspective, we tested the capacity of available phylogenetic methods to detect the patterns of diversity dependence predicted by our model. Phylogenetic analyses of diversification now routinely incorporate diversity-dependent factors[10,42,47], yet it has been unclear if and why diversity-dependent models are superior to alternatives[45,48–50]. During the first stage of the radiation, the rates of speciation and extinction are constant, but they are correctly identified as diversity-independent in only half of the simulation replicates (Supplementary Fig. 11a–f). The initial radiation ends when the number of species reaches 9; this low diversity may result in a low power to discriminate between diversity-independent and diversity-dependent rates. Although constant in both phases 1 and 2, the speciation rate is usually lower during the niche self-structuring phase (stage 2). When phylogenies are analyzed at the end of stage 2, although a diversity-independent model with one shift of the evolutionary rates is always better than a diversity-independent model with rates constant through time (not shown), the shift in the speciation rate between phases 1 and 2 appears to be detected as a signature of negative diversity-dependence by phylogenetic analyses (Supplementary Fig. 11g–l). This may be due to a confounding effect of the extinction rate which rises markedly as diversity increases from stage 1 to stage 2. In the stationary phase (diversification stage 3), the speciation rate is almost always correctly identified as negatively diversity-dependent (Supplementary Fig. 11m–r).

Our results call for the development of new phylogenetic models that distinguish between different temporal stages of diversity-dependence and diversity-independence. These models would make it possible to identify the different stages of diversification in a clade's history; reveal the corresponding pattern of speciation and extinction rates across these stages; and test the

model predictions regarding the ecological, genetic, and environmental controls of macroevolutionary rates in each stage.

## Methods

To link macroevolutionary dynamics to organismal, ecological, and environmental processes, we used a refined version of the individual-based eco-evolutionary model of species diversification introduced in Aguilée et al.[19], Aguilée et al.[21], and Gascuel et al.[23]. The model scales up from local ecological interactions and individual dispersal, integrates the evolution of prezygotic and postzygotic mechanisms of reproductive isolation, and represents geographical events on a slow timescale as an alternation of geographic isolation and contact between populations. Heritable traits of individuals, such as body size, influence their life history and ecological interactions and thus shape the ecological state of the community, which in turn generates selection driving trait evolution. The eco-evolutionary feedback between the population trait distribution and the state of the ecological system drives phenotypic evolution. Emerging eco-evolutionary theory[51] emphasizes that phenotypic diversification itself can mold the ecological niches of interacting species[30,52–56]; our model is designed to capture how feedbacks between ecological and evolutionary processes interact with geographic and other large-scale abiotic factors to shape speciation and extinction rates over a clade's history.

**Model construction**. General assumptions about individuals and their environment: We model the macroevolutionary dynamics of a clade in a dynamic two-dimensional landscape made up of $n \times n$ sites. As the clade diversifies, species assemble in a metacommunity distributed across this landscape. Environmental resources vary in both dimensions: each site is characterized by a different environmental optimum, $z^* = (x_1^*, x_2^*)$. In the baseline version of the model, environmental optima follow a two-dimensional gradient, with $x_1^*$ and $x_2^*$ increasing by a constant amount $\delta_x$ between adjacent sites in either direction (Fig. 1a and Supplementary Fig. 1). The robustness of results derived from the resource-gradient scenario is tested by randomizing environmental optima across geographical sites (Supplementary Fig. 1). The landscape is dynamic as geographical barriers can arise between adjacent sites (at rate $f$), thereby preventing migration and gene flow, or disappear (at rate $c$), allowing migration at birth as in the island model, with equal probability for the offspring to reach any of the connected sites.

The "ecological phenotype" of individual organisms is determined by their resource utilization strategy, described by two quantitative traits measured along each environmental dimension, and thus denoted by $x_1$ and $x_2$. Traits $x_1$ and $x_2$ are assumed to be genetically variable; they are determined by $L_k(k = x_1, x_2)$ unlinked additive quantitative loci on autosomal chromosomes with no environmental effect. Groups of phenotypically similar individuals inhabiting the same geographical site form subpopulations; subpopulations are grouped into species according to their level of reproductive isolation (see Supplementary Fig. 2). Reproductive isolation between subpopulations can result from two different mechanisms: reversible prezygotic reproductive isolation, resulting from the joint evolution of assortative mating and phenotypic traits; or genetic incompatibility which may result from genetic drift in allopatry. In subsequent sections, we describe how each mechanism is incorporated in the model. Here "population" refers to the set of all subpopulations of the same species coexisting in a given site.

As the landscape dynamics unfold, successive rounds of disconnection and connection of sites lead to repeated phases of allopatry and sympatry between local communities. Physical barriers favor phenotypic divergence between populations, and hence create opportunities for allopatric speciation, whereas secondary contacts set the stage for character displacement and reinforcement. The history of species origination and species loss thus emerges from the joint operation of biotic processes operating at the level of individuals, and abiotic processes operating at the level of sites. Note that the polygenic trait inheritance that our model assumes impedes sympatric speciation due to disruptive selection at the environmental optima[28,29].

Reproductive isolation and assortative mating: To model reproductive isolation resulting from the evolution of assortative mating, we consider "magic traits for speciation" driving ecological speciation[57–60]. To compute the mating probability between two individuals, we introduce an additional phenotypic component, the "choosiness trait", $a$, which measures the degree to which ecological similarity influences the probability of mating. Like the ecological traits $x_1$ and $x_2$, the choosiness trait $a$ is polygenic and determined by $L_a = 16$ unlinked additive loci. If $a > 0$, mating is favored by ecological similarity; if $a < 0$, mating is favored by ecological dissimilarity. The mating probability between individuals $i$ and $j$ is computed as

$$Q(i,j) = \begin{cases} \left(1 - \frac{1}{2}e^{-a_i^2}\right)e^{\left(-\frac{||z_i - z_j||^2}{(2/(a_i^2 c_{am})^2)}\right)} & \text{if } a_i > 0 \\ 0.5 & \text{if } a_i = 0 \\ 1 - \left(1 - \frac{1}{2}e^{-a_i^2}\right)e^{\left(-\frac{||z_i - z_j||^2}{(2/(a_i^2 c_{am})^2)}\right)} & \text{if } a_i < 0 \end{cases}$$

with $|a_i| > |a_j|$ (meaning that we use the choosiness trait value of the choosiest individual). Here, $c_{am}$ is a scaling constant, $z_i$ is the ecological phenotype ($x_1, x_2$) of

individual $i$, and $a_i$ is the assortative mating trait of individual $i$. This Gaussian mating function has the minimal biological realism required[19,21,23]: it is a continuous function in $a_i$, individual $i$ has no preference when $a_i = 0$ and mates assortatively (resp. disassortatively) when $a_i > 0$ (resp. $a_i < 0$), and choosiness increases when $|a_i|$ increases.

For any two subpopulations, we assess prezygotic isolation by computing an average cross-breeding probability using the mating function above with the average phenotypic traits of the subpopulations; if this probability is above a certain threshold, that we call "assortative mating threshold" (AMT, set to AMT = 0.01 by default; Supplementary Table 1), and then the two subpopulations are considered not to be reproductively isolated. Accounting for these average trait values in each subpopulation (instead of considering mating probabilities between all pairs of individuals) is an approximation, but it has no impact on the results while significantly reducing computation time[23].

Reproductive isolation and genetic incompatibilities: Genetic incompatibilities that build up between subpopulations through mutation can lead to irreversible postzygotic reproductive isolation[24–27]. Genetic incompatibilities are controlled by a set of loci different from the loci controlling the ecological and mating traits. Postzygotic reproductive isolation due to genetic incompatibilities may thus evolve independently of prezygotic reproductive isolation due to assortative mating. We compute the genetic distance between two potential mates by recording all the loci controlling incompatibilities carrying different alleles. We use a "genetic incompatibility threshold" (GIT) to set the genetic distance above which two individuals cannot mate.

We proceed similarly at the level of subpopulations: we compute the genetic distance between each pair of subpopulations by recording all the loci controlling incompatibilities carrying divergent alleles in these subpopulations, and by weighting each of these loci by its frequency in the subpopulation. We consider the subpopulations to be reproductively isolated when the genetic distance between the subpopulation is above the GIT value. The GIT value was set to 15 by default (Supplementary Table 1). This value was chosen to ensure that (Supplementary Fig. 7)

- The genetic incompatibility route to speciation did not obscure the effect of ecological factors (competition width and local resource abundance) on macroevolutionary rates;
- Irreversible species reproductive isolation occurring on the long term would prevent hybridization between species that were old and geographically isolated, but had converged to similar phenotypes;
- Mutations responsible for genetic incompatibility were not selected against (which is the case as long as GIT > 1).

Life cycle and competition: Reproduction is sexual and individuals reproduce at a constant rate, $r$. When a reproduction event takes place, an individual is selected at random and its potential mate is chosen randomly among the individuals of the opposite sex inhabiting the same geographical site. The potential mates reproduce only if they are not reproductively isolated, either prezygotically (due to assortative mating) or postzygotically (due to genetic incompatibility); this is done exactly as to assess the reproductive isolation between pairs of populations, as explained in previous sections. If the pair of individuals is reproductively isolated, then another potential mate is drawn at random and the process is repeated until reproduction occurs, or a maximum number (set to 50) of potential mates have been tried. The cost of choosiness is thus assumed to be small[61,62], which facilitates the evolution of assortative mating. When reproduction occurs, the offspring sex is determined randomly assuming a balanced sex-ratio.

The offspring ecological and choosiness traits are determined from the random independent segregation of the $L_k$ parental loci that code for each trait $k$ ($k = x_1, x_2,$ or $a$). Mutation occurs at each of these loci with probability $\mu_p$ and the mutant allelic value is drawn from a normal distribution with mean equal to the parental allelic value and with standard deviation $s_k\sqrt{2L_k}$. This mutation size at the allele level results in a mutational variance $s_k^2$ at the level of trait $k$, regardless of the number of loci[63]. Finally, the offspring inherits loci harboring incompatible alleles from its parents following a random independent segregation of parental loci, and may acquire a new incompatible allele according to an infinite site model, with mutation probability $\mu_n$.

The individual death rate depends on the local (i.e., within the same geographical site) carrying capacity and on the strength of local competition: individual $i$ with ecological phenotype $z_i = (x_1, x_2)$ dies at rate $d(z_i) = \frac{r}{K(z_i)}\sum_{j=1, j \neq i}^{n} C(z_i, z_j)$, where $n$ is the number of individuals inhabiting the same geographical site, $K$ is the carrying capacity function, $r$ is the per-capita constant birth rate, and $C$ is the competition kernel. This framework models competition for resources as a process that is both density-dependent[64] and frequency-dependent (i.e., stronger between individuals with more similar ecological traits)[65,66]. The carrying capacity function models a continuous distribution of resources within each geographical site; we assume that it has a Gaussian shape with maximum $K^*$ at phenotype $z^* z^* = (x_1^*, x_2^*)$, and standard deviation $\sigma_K$. Thus, $z^*$ is the environmental optimum of the geographical site, and $K(z_i) = K^* e^{\left(-\frac{||z_i - z^*||^2}{2\sigma_K^2}\right)}$. The competition kernel is also Gaussian, with a maximum value of 1 when individuals ($i, j$) have identical ecological

phenotypes and standard deviation $\sigma_C$, hence

$$C\left(z_i, z_j\right) = e^{\left(-\frac{||z_i - z_j||^2}{2\sigma_C^2}\right)}$$

To favor phenotypic diversification in each site, we set the standard deviation of the carrying capacity function, $\sigma_K$, to be larger than that of the competition kernel, $\sigma_C$[52,65,67,68]. Gaussian kernels have been widely used to model competition, but are also criticized for introducing structural instability in deterministic models of resource use evolution[69–71]. Structurally unstable outcomes are unlikely in our model, because multiple sources of stochasticity are included (in life history events, mutation, dispersal, and landscape dynamics).

**Numerical simulations**. Diversification is initiated from one ancestral species spread over all geographical sites (150 individuals per site). Initial allelic values at the loci of the ecological and choosiness traits $(x_1, x_2, a)$ are randomly chosen in a Gaussian distribution centered at zero. The ancestral species is therefore adapted to the geographical site with intermediate environmental optimum $\left(x_1^* = 0, \ x_2^* = 0\right)$. This prevents border effects in the initiation of diversification, and captures a range of empirical situations at the onset of diversification, such as the colonization of new geographical areas with few ecological competitors, or the evolution of a key innovation[39,40,57,72,73].

The default parameter values used in this study are given in Supplementary Table 1. Simulations of species diversification are typically run for 100,000 generations, which ensures that the macroevolutionary dynamics reach a stationary state. Diversification is therefore much more rapid than typically observed in adaptive radiations[57]; this is primarily due to high mutation rates (Supplementary Table 1), similar to those used in other individual-based models of diversification[21,34,65,74]. Such high mutation rates help reduce computation time without affecting the patterns of variation in macroevolutionary rates. These patterns depend on the rate of phenotypic trait evolution, and on how the latter scales with the rate of landscape dynamics[19,21,75]. Thus, we expect macroevolutionary rate patterns to be invariant under smaller mutation rates and higher expected phenotypic variance (e.g., with $\mu_k = 10^{-5}$, $s_z^2 = 0.01$, $s_a^2 = 0.04$, and $L_k = 6$; see Aguilée et al.[21]); and under smaller mutation rates along with smaller rates of landscape dynamics, on a longer timescale.

We monitored the evolutionary history of species through time by extracting the composition of the metacommunity every 100 generations (as presented in Supplementary Fig. 2), and used these data to compute (per-species) speciation and (per-species) extinction rates, and to build the phylogenetic trees of extant species at the end of each stage of the radiation. It was previously checked[23] that similar relationships between species are obtained with a sampling interval of ten generations. We analyzed the dependence of macroevolutionary rates on species diversity by measuring the expected waiting time before a speciation (respectively extinction) event as a function of extant diversity. A general pattern of diversification in three major stages emerged from our simulations (Fig. 1). To characterize and understand this pattern, we calculated and analyzed the rate of species diversification (speciation rate–extinction rate), the macroecological patterns (metacommunity size and species-abundance distributions), the species' age distribution during each stage of diversification, and the rate of species turnover at the stationary state (at which the speciation rate and the extinction rate are equal). The rate of diversification in stage 2 and species diversity at the stationary state were evaluated numerically by fitting a von Bertalanffy growth curve to species diversity, $N$, through time $t$, starting at $N = 9$ at the beginning of stage 2:

$$N(t) = a\left(1 - be^{-ct}\right)^3$$

We investigated the effect of three biotic factors (genetic incompatibility, resource abundance, and competition width) and three abiotic factors (pace of landscape dynamics, mean site isolation time, and local catastrophe rate) on macroevolutionary dynamics.

- To analyze the effect of genetic incompatibility, we tested the values of GIT ranging from 4 to 25 (the default value being GIT = 15; Supplementary Table 1). The rate of accumulation of loci responsible for genetic incompatibilities was kept constant, equal to $\mu_n = 10^{-3}$.
- To analyze the effect of competition, we defined the ("scaled") competition width as the range of resources utilized by each species relatively to the total range of available resources in each site, which is inversely proportional to the number of species that may coexist in this site. It is measured by $\sigma_C/\sigma_K$[66]. We tested the values of $\sigma_C/\sigma_K$ ranging from 0.25 to 0.75 (the default value being $\sigma_C/\sigma_K = 0.4$, with $\sigma_C = 0.4$, $\sigma_K = 1.0$; Supplementary Table 1).
- Resource abundance is measured by the local carrying capacity, $K^*$, which determines how many individuals with phenotypes at the environmental optimum can coexist in a given site. We tested values of $K^*$ ranging from 75 to 375 individuals (the default value being $K^* = 150$; Supplementary Table 1).

- The pace of landscape dynamics (cycles of connection and disconnection of adjacent sites) is measured by $1/\left(\frac{1}{f} + \frac{1}{c}\right)$, where $f$ is the rate of barrier arising and $c$ is the rate of barrier removal. We performed simulations with the pace of landscape dynamics ranging from $9.9 \times 10^{-6}$ to $1.4 \times 10^{-4}$ (the default pace being $4.76 \times 10^{-5}$, with $f = 10^{-3}$, $c = 5 \times 10^{-5}$; Supplementary Table 1).
- Time in isolation is measured by the fraction of time that a barrier is raised on average over the long term, which is equal to $f/(f + c)$. When the time in isolation is long, species in adjacent geographical sites tend to spend more time in allopatry, and less time in sympatry. We tested the values of isolation time ranging between 52% and 99% (the default value being 95%; Supplementary Table 1). We used the values of $f$ and $c$ which ensured that the pace of landscape dynamics was kept constant while changing isolation time, and vise-versa.
- Local catastrophes strike sites at random and cause immediate extinction of all populations present in the site. The rate of local catastrophe, denoted by $cr$, was increased up to $5 \times 10^{-4}$ (it was set to zero by default; Supplementary Table 1).

Note that under high resource abundance and slow pace of landscape dynamics, simulations were run over 200,000 or 300,000 generations (instead of 100,000 generations by default) to ensure convergence to stationary numbers of individuals and species. Our results for each parameter set are based on 50 simulation replicates.

To explain the effect (or lack thereof) of biotic and abiotic factors on speciation and extinction rates, we focused on key ecological and genetic variables that potentially mediate these effects:

- The intensity of interspecific competition, which is defined for any given species as the average, across its $n_i$ individuals, of the competition felt by each of its individual $i$ due to the presence of $n_j$ individuals $j$ of other species sharing in the same geographical site(s), is scaled by the local carrying capacity. For each individual $i$, this interspecific competition is measured as $\frac{1}{K^*} \sum_{j=1}^{n_j} C(z_i, z_j)$.
- The average phenotypic divergence of any given species from its parental species, is measured as the Euclidian distance between the mean phenotype of each species. Phenotypic divergence influences competition intensity and hybridization risk, either at present or at future secondary contact.
- The average assortative mating trait $a$.
- The average degree of populations' maladaptation relative to the local environmental optimum $z^*$. This is measured by the average squared distance between the optimum and all individual phenotypes in a given population, averaged over all populations in the metacommunity.
- The average species' geographic range size. This is the number of sites occupied by each species, averaged over all species in the metacommunity.
- The population abundance (average species' abundance per site).
- The risk of hybridization, which is measured as the maximum hybridization probability $Q(i, j)$ (see the section "Reproductive isolation and assortative mating") between a subpopulation $i$ of any given species and a subpopulation $j$ of another species––located either in the same or in another geographical site, conditioned on both subpopulations not being reproductively isolated by genetic incompatibilities.

To evaluate the influence of the above variables on speciation and extinction rates, we collected these data for all species combined, or specifically for species about to speciate (i.e., undergoing speciation within the next 200 generations), or species about to go extinct (i.e., undergoing extinction within the next 200 generations), or "static species", which do not speciate or go extinct within the next 2000 generations.

**Data availability**. The computer code of the simulations and of the analyses is provided as Supplementary Data 1. All relevant data are available from the authors.

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

## Acknowledgements

We thank Michael S. Barker, Florence Débarre, and Michael Sanderson for discussion. We are grateful to the Institut National de la Recherche Agronomique (INRA) for giving us access to the Migale bioinformatic platform and its high-performance computing cluster. Part of the analyses were run on the cluster EDB-Calc (which uses software developed by Rocks Cluster Group at the San Diego Supercomputer Center, University of California) of the laboratory Evolution et Diversité Biologique. We thank Pierre Solbès for support with the EDB-Calc system. Financial support was received from the Center for Interdisciplinary Research in Biology (Collège de France), the Centre National de la Recherche Scientifique (Mission pour l'Interdisciplinarité, Pépinière de site PSL "Eco-Evo-Devo"), France Laboratories of Excellence "TULIP" (PIA-10-LABX-41) and "MemoLife" (PIA-10-LABX-54), and the Partner University Fund for ENS-University of Arizona cooperation.

## Author contributions

R.F. and A.L. directed the study. R.A., F.G., and A.L. designed and implemented the model. F.G. ran the simulations and analyzed the data with the contribution of R.A. All authors interpreted the results. R.F. and F.G. wrote the first versions of the manuscript. All authors contributed to the final version of the paper.

## Additional information

**Competing interests:** The authors declare no competing interests.

