## [Peer Review File · Nature Communications]

Reviewers' comments:

Reviewer #1 (Remarks to the Author):

First, I want to congratulate the authors - this is a truly impressive work. I'd say this is the most significant advance in theoretical speciation research in the last 10 or so years. That they managed to bring so many different things (local adaptation, competition, incompatibilities, selection gradients, transient geographic barriers, phylogenies, macroecological patterns) in a single modeling framework and get meaningful biological insights out of this is amazing. The topics they focus are of great current interest and importance. Their approach is not just novel but groundbreaking. Their conclusions make perfect sense. There is a wealth of information, explanations, and predictions here which I am sure will inspire and motivate many empirical biologists. The paper is also very well written (and provides a very good overview of scientific background). It is also quite well appropriate for this journal and its audience. My comments below are relatively minor.

lines 87-88: your model also includes post-zygotic isolation in the form of selection against maladaptive hybrids

Fig.3: not all variables of the vertical axes appear to be explicitly defined (unless I missed that).

454: say explicitly that all loci are unlinked.

sections 1.2 and 1.3 of Methods are a bit confusing. I'd recommend to define breeding probabilities for pairs of *_individuals_* first and then say that in identifying species you just use the average values for computation simplicity.

484: the expression for Q requires an explanation and perhaps a simplification of notation (e.g. why do you need both u_i and a_i there, why two different exponential terms?).

496: say explicitly that these are the same loci as those controlling the ecological and mating traits.

524: isn't there a typo in the expression for the standard deviation?

Sergey Gavrillets

Reviewer #2 (Remarks to the Author):

This paper presents a simulation study of adaptive radiations in models for resource competition in geographically structured populations. In these models, speciation first occurs allopatrically, with new lineages moving into previously unoccupied geographical areas that select for different optimal phenotypes. Speciation occurs through postzygotic isolation (which happens by way of the assumptions in the model). In a second phase, when all geographical areas are occupied, speciation occurs in sympatry owing to competition and prezygotic isolation, and speciation and extinction rates become more or less constant.

It is interesting to see these events unfold in a single model, but none of this is either novel or surprising. Colonization of empty geographical niches with phenotypic divergence due to local adaptation is how most evolutionary biologists envision radiations, and speciation due to competition for resources has been studied quite extensively both theoretically and empirically. It is curious that the authors call the second phase of speciation "disadaptive" diversification, despite the fact that, according to the authors' conceptual backdrop for the paper, this is the mode of speciation that Darwin had in mind. Also, this mode of speciation has been called "adaptive

speciation" in the past (see e.g. the book by Dieckman et al 2004), ostensibly because the species move away from the environmental optimum as an adaptation to avoid competition.

The paper contains a number of quantitative refinements of the general picture, such as the reported "speciation slowdown" (which has been repeatedly predicted elsewhere and is due to restriction of species ranges as niches fill up), and how various parameters such as rates of catastrophes, resource abundance and changes in the landscape affect the level of diversity in the metacommunity. However, overall this paper appears to be better suited for a more specialized journal.

Reviewer #3 (Remarks to the Author):

This paper presents a simulation model of speciation and extinction in a subdivided landscape where each site has a different resource availability in a two-dimensional continuous niche space. The approach here is to build a fairly complex individual based simulation to address what role Darwin's "principle of divergence" (PoD) plays in regulating speciation and extinction rates of a clade on a landscape. The main conclusions are that the initial diversification is little affected by the principle of divergence but it can play a major role in later stages of diversification, especially in the form of incipient species that are insufficiently diverged from their ancestral species going extinct at a higher rate. Nonetheless, "abiotic" factors such as the overall productivity of the environment and the fluctuation rate determine the overall species richness and speciation and extinction rates.

I find the paper to be interesting, and a potentially valuable contribution to the currently active theoretical literature on how macro-evolutionary rates are affected by microevolutionary and ecological dynamics. That said, I have some concerns about the overall framing and the generalizability of some conclusions to changes in model parameters.

First off, the framing of the paper in the introduction to my mind was a bit odd. In particular, the authors claim that the PoD was rejected by the modern synthesis. Perhaps it was in the speciation literature, but in community ecology, PoD has remained very much alive, and the closely related (if not identical) concept of limiting similarity forms the mainstay of ecological theories of diversity. Ignoring this thread I think detracts from the framing. More substantively, the arguments in this paper resemble some of the early mathematical niche evolution literature (precursors to adaptive dynamics) by MacArthur, Roughgarden, Abrams, et al. between the 60's and 80's, which are all based on limiting similarity and competition as their central concept. Those models made quite a few simplifying assumptions (e.g., no modeling of sexual reproduction) but in many cases also got analytical results, and those results seem consistent with the current simulation. The authors do cite work from the more recent adaptive dynamics literature from the 2000s, but I think the framing in the introduction is still a bit misleading. I wouldn't try to pitch the paper as "rescuing" a mistakenly discarded principle of Darwin.

Coming to the substance of the model, I find the construction to be largely intuitive. The results also seem to make sense, but I am somewhat skeptical of whether they will generalize beyond the specific set up considered here. Before launching into my skepticism, I should say that the authors already do quite a bit of sensitivity analyses to various parameters, which deserves credit.

Reading the model, I had two main concerns: first, I think I understand why the authors choose a very regular landscape with 9 sites, distributed in a lattice along two resource axes (namely, they wanted to keep the resource environment fixed). There is some overlap between the sites' environmental niches: the differences between the optima and the standard deviation of the within-site resource distribution are set to the same value. But I feel that their results can potentially change if this overlap and/or the distribution of sites over the resource landscape is changed. For example, with less overlap, the effect of fragmentation (provided it's high enough)

might not be as strong, since locally adapted species will have a hard time to colonize other sites to begin with. I would feel more confident in the conclusions of the paper if the degree of overlap between sites was explored further.

Another factor that I think can substantially change results is something authors note (more or less in passing) in the discussion: that they do not allow the evolution of niche breadth. I feel that this is a potentially serious limitation: as the authors note, evolution of generalists and specialists can change overall diversity and species ranges substantially. I am sympathetic to the argument that they left this out to keep the model simple, but I believe that some of the specific conclusions might be substantially altered if one doesn't make this simplifying assumption, which somewhat detracts from the generality of the conclusions. For example, it might be expected that intuitively, that (early) evolution of generalists can "choke" adaptive diversification by making the ephemeral speciation more common in early stages of diversification. This will probably depend both on the breadth of local resource distributions and connectivity (as well as the overlap). But in any case, it raises questions in my mind about how robust the conclusions from the model are to variations in model structure.

On a slightly different note, I believe that the authors can do more to be more specific about their conclusions. Currently the discussion presents a somewhat muddled message. One suggestion I'd make on this front would be, to use current phylogenetic methods on the simulated phylogenies, to test how different methods (e.g., w/ and w/o density dependent speciation and extinction rates) fare in a model where they know all the ingredients. That would put the discussion on a more solid footing (allowing the authors to make sharper statements), and serve as a validation of these methods (or not) in a model with known rates.

Related to the that, a clarification question: it wasn't obvious to me, in the description of species delimitation in the model, what the time-steps were. Are the authors sampling the populations in each time-step (after each reproduction event) or do they coarse-grain in time? Seems like that might affect the phylogenetic relationships being inferred.

Overall, I believe this is a potentially positive contribution, but would benefit from a slightly less grandiose framing as well as some robustness checks of the simulation model.

Manuscript #17-21576 initially entitled “Darwin’s principle of divergence and the controls of macroevolutionary rates”, by R. Aguilée, F. Gascuel, A. Lambert and R. Ferriere

New title: “How ecological competition, genetic divergence, and landscape dynamics control the macroevolutionary rates of speciation and extinction”

Response to Reviewers

Response to Reviewer #1 (Remarks to the Author):

First, I want to congratulate the authors - this is a truly impressive work. I'd say this is the most significant advance in theoretical speciation research in the last 10 or so years. That they managed to bring so many different things (local adaptation, competition, incompatibilities, selection gradients, transient geographic barriers, phylogenies, macroecological patterns) in a single modeling framework and get meaningful biological insights out of this is amazing. The topics they focus are of great current interest and importance. Their approach is not just novel but groundbreaking. Their conclusions make perfect sense. There is a wealth of information, explanations, and predictions here which I am sure will inspire and motivate many empirical biologists. The paper is also very well written (and provides a very good overview of scientific background). It is also quite well appropriate for this journal and its audience. My comments below are relatively minor.

- We thank Dr. Gavrilets for his positive feedback.

lines 87-88: your model also includes post-zygotic isolation in the form of selection against maladaptive hybrids

- This point has been included, as suggested (lines 95-98).

Fig.3: not all variables of the vertical axes appear to be explicitly defined (unless I missed that).

- These variables were explicitly defined in the section Methods, subsection Numerical simulations. We have now added a reference to this section in the figure caption, so that readers can more easily find the definition of each variable (lines 636-637).

454: say explicitly that all loci are unlinked.

- Done (lines 704 and 728).

sections 1.2 and 1.3 of Methods are a bit confusing. I'd recommend to define breeding probabilities for pairs of _individuals_ first and then say that in identifying species you just use the average values for computation simplicity.

- As suggested by the reviewer, we have rewritten these sections. We now define the processes at the individual level and then we explain how individual rules are translated at the population

level (lines 722-764).

484: *the expression for Q requires an explanation and perhaps a simplification of notation (e.g. why do you need both u_i and a_i there, why two different exponential terms?).*

- We simplified the notation and added an explanation of the properties of the mating function (lines 729-735).

496: *say explicitly that these are the same loci as those controlling the ecological and mating traits.*

- The set of loci controlling genetic incompatibilities and the two sets of loci controlling the ecological and mating traits are different. Post-zygotic reproductive isolation due to genetic incompatibilities may thus evolve without any ecological divergence, and conversely a large ecological divergence does not necessarily prevents hybrids to be produced. We added these explanations in the revised manuscript (lines 746-751).

524: *isn't there a typo in the expression for the standard deviation?*

- There was no typo in the expression, but our notation $\sigma_{\mu_k}^2$ for the mutational variance was misleading given that μ_k is defined as the mutation rate. We changed the notation $\sigma_{\mu_k}^2$ into s_k^2 to avoid any confusion (lines 779-783).

Response to Reviewer #2 (Remarks to the Author):

This paper presents a simulation study of adaptive radiations in models for resource competition in geographically structured populations. In these models, speciation first occurs allopatrically, with new lineages moving into previously unoccupied geographical areas that select for different optimal phenotypes. Speciation occurs through postzygotic isolation (which happens by way of the assumptions in the model). In a second phase, when all geographical areas are occupied, speciation occurs in sympatry owing to competition and prezygotic isolation, and speciation and extinction rates become more or less constant.

It is interesting to see these events unfold in a single model, but none of this is either novel or surprising. Colonization of empty geographical niches with phenotypic divergence due to local adaptation is how most evolutionary biologists envision radiations, and speciation due to competition for resources has been studied quite extensively both theoretically and empirically.

- We agree that each part of the diversification process has been previously extensively studied. What is novel and unexpected is how different mechanisms and factors of the diversification process combine and shape macroevolutionary rates. We have largely rewritten the Abstract, the Introduction, parts of the Results and of the Discussion to better emphasize the novelty of our study.

It is curious that the authors call the second phase of speciation "disadaptive" diversification, despite the fact that, according to the authors' conceptual backdrop for the paper, this is the mode of

speciation that Darwin had in mind. Also, this mode of speciation has been called “adaptive speciation” in the past (see e.g. the book by Dieckman et al 2004), ostensibly because the species move away from the environmental optimum as an adaptation to avoid competition.

- We used the term “disadaptive” to emphasize that during the second phase of the radiation, species ecological traits evolve away from the environmental optimum of each site. We agree with the risk of confusion that the term may generate, for the reasons given by the Reviewer. We therefore avoided the use of “disadaptive” in our revised manuscript.

The paper contains a number of quantitative refinements of the general picture, such as the reported “speciation slowdown” (which has been repeatedly predicted elsewhere and is due to restriction of species ranges as niches fill up), and how various parameters such as rates of catastrophes, resource abundance and changes in the landscape affect the level of diversity in the metacommunity. However, overall this paper appears to be better suited for a more specialized journal.

- The focus of our work is on explaining variation in macroevolutionary rates. To our knowledge, this is the first model that builds on individual-level mechanistic processes to scale up and predict patterns of macroevolutionary rates and their variation. We believe there is general interest from a broad community of evolutionary biologists and ecologists to better understand how macroevolutionary rates of speciation and extinction are controlled. Recent review articles, such as Weber et al. 2017 in *Trends in Ecology and Evolution*, make a clear call for studies like ours.

The reviewer’s comments suggested to us that the Abstract, Introduction, and Discussion were in need of a thorough rewrite, so that the novelty and significance of our results would be more easily appreciated. This is what we have done. See also below, our response to Reviewer #3’s comment on using Darwin’ principle of divergence to frame the study.

Response to Reviewer #3 (Remarks to the Author):

This paper presents a simulation model of speciation and extinction in a subdivided landscape where each site has a different resource availability in a two-dimensional continuous niche space. The approach here is to build a fairly complex individual based simulation to address what role Darwin’s “principle of divergence” (PoD) plays in regulating speciation and extinction rates of a clade on a landscape. The main conclusions are that the initial diversification is little affected by the principle of divergence but it can play a major role in later stages of diversification, especially in the form of incipient species that are insufficiently diverged from their ancestral species going extinct at a higher rate. Nonetheless, “abiotic” factors such as the overall productivity of the environment and the fluctuation rate determine the overall species richness and speciation and extinction rates.

I find the paper to be interesting, and a potentially valuable contribution to the currently active theoretical literature on how macro-evolutionary rates are affected by microevolutionary and ecological dynamics. That said, I have some concerns about the overall framing and the generalizability of some conclusions to changes in model parameters.

First off, the framing of the paper in the introduction to my mind was a bit odd. In particular, the authors claim that the PoD was rejected by the modern synthesis. Perhaps it was in the speciation

literature, but in community ecology, PoD has remained very much alive, and the closely related (if not identical) concept of limiting similarity forms the mainstay of ecological theories of diversity. Ignoring this thread I think detracts from the framing. More substantively, the arguments in this paper resemble some of the early mathematical niche evolution literature (precursors to adaptive dynamics) by MacArthur, Roughgarden, Abrams, et al. between the 60's and 80's, which are all based on limiting similarity and competition as their central concept. Those models made quite a few simplifying assumptions (e.g., no modeling of sexual reproduction) but in many cases also got analytical results, and those results seem consistent with the current simulation. The authors do cite work from the more recent adaptive dynamics literature from the 2000s, but I think the framing in the introduction is still a bit misleading. I wouldn't try to pitch the paper as "rescuing" a mistakenly discarded principle of Darwin.

- As emphasized by the Reviewer, the ecological literature is rich in theory and models that relate competition, niche evolution and diversity. However, as also alluded to by the Reviewer, early ecological niche theory and contemporary theory of ecology-driven trait evolutionary diversification could not say much about the macroevolutionary process, because these theories are generally oblivious to the genetics of speciation. Precisely, our goal was to incorporate simple genetic processes of reproductive isolation in order mechanistically to link ecological trait evolution and the origination and extinction of biological species. Our results show that ecological processes do shape macroevolutionary patterns and influence the macroevolutionary rates provided that genetical mechanisms of reproductive isolation are explicitly taken into account.

To echo the reviewer's unease with our framing of the study, we largely rewrote the Abstract and Introduction, which have now a clear and direct focus on understanding the biotic and abiotic drivers of macroevolutionary rates. Darwin's principle of divergence is now only addressed in the Discussion (lines 363-382).

Coming to the substance of the model, I find the construction to be largely intuitive. The results also seem to make sense, but I am somewhat skeptical of whether they will generalize beyond the specific set up considered here. Before launching into my skepticism, I should say that the authors already do quite a bit of sensitivity analyses to various parameters, which deserves credit.

Reading the model, I had two main concerns: first, I think I understand why the authors choose a very regular landscape with 9 sites, distributed in a lattice along two resource axes (namely, they wanted to keep the resource environment fixed). There is some overlap between the sites' environmental niches: the differences between the optima and the standard deviation of the within-site resource distribution are set to the same value. But I feel that their results can potentially change if this overlap and/or the distribution of sites over the resource landscape is changed. For example, with less overlap, the effect of fragmentation (provided it's high enough) might not be as strong, since locally adapted species will have a hard time to colonize other sites to begin with. I would feel more confident in the conclusions of the paper if the degree of overlap between sites was explored further.

- Our results are robust to at least some variation in the landscape structure. The number of sites we used was a compromise between landscape complexity and the model's computational tractability. Since it was computationally challenging to increase the number of sites further, we explored the consequence of *reducing* the number of sites. Fewer sites tend to reduce the number of species that can evolve, but does not alter the qualitative pattern of diversification

that we report in the manuscript. Rather than adding to the current manuscript (which is already substantially above the page limit) we referred to our earlier study (Aguilée et al. 2013 *Evolution*) where we used a landscape consisting of a central site surrounded by 5 other sites, with only 3 environmental optima. Diversification patterns shown in Aguilée et al. (2013) are consistent with those reported here. With more sites (9) and more environmental optima (9) than in Aguilée et al.'s model, we obtained a broader range of diversification dynamics, which made our investigation of the determinants of macroevolutionary rates possible.

Regarding the degree of resource overlap between adjacent sites, there is little effect on the diversification process, as long as the overlap is neither too small nor too large. On the one hand, resource overlap should not be too large; otherwise allopatric divergence when sites are disconnected would be too weak to allow reinforcement at secondary contact, thus preventing new species to form. On the other hand, resource overlap should be large enough to prevent stochastic extinction, due to low population size, of phenotypically similar subpopulations in adjacent sites with different optima. In other words, resource overlap should be sufficient to allow species to spread from sites to sites. In our revision, we referred to the results with respect to the level of resource overlap that Pontarp *et al.* reported previously (2012 *Evol. Ecol. Res.*; 2015 *Am. Nat.*). They found that more diversification occurred for low to intermediate resource overlap. (Note that in Pontarp *et al.* (2012, 2015), sympatric speciation is not prevented by polygenic trait inheritance which explains that diversification is limited but still possible when resource overlap is large.)

In the revised manuscript, the Discussion section now includes a separate paragraph (lines 334-362) in which we highlight the model assumptions (including the degree of resource overlap) that are critical for the general pattern of diversification predicted by our model.

Another factor that I think can substantially change results is something authors note (more or less in passing) in the discussion: that they do not allow the evolution of niche breadth. I feel that this is a potentially serious limitation: as the authors note, evolution of generalists and specialists can change overall diversity and species ranges substantially. I am sympathetic to the argument that they left this out to keep the model simple, but I believe that some of the specific conclusions might be substantially altered if one doesn't make this simplifying assumption, which somewhat detracts from the generality of the conclusions. For example, it might be expected that intuitively, that (early) evolution of generalists can "choke" adaptive diversification by making the ephemeral speciation more common in early stages of diversification. This will probably depend both on the breadth of local resource distributions and connectivity (as well as the overlap). But in any case, it raises questions in my mind about how robust the conclusions from the model are to variations in model structure.

- In this study we deliberately kept niche width constant to ensure that the evolution of specialization does *not* contribute to diversification. This is stated as a key assumption early on in the manuscript, in the model description (lines 104-107): "By assuming constant competition width, we control for the effect that evolving specialization may have on diversification. In other words, keeping competition width constant ensures that the evolution of specialization does *not* contribute to the diversification of the clade." How niche width co-evolves with niche position (the two ecological traits in this model) requires new computational developments that are beyond the scope of this study. However, re-examining our results in the light of existing literature yields qualitative insights into the putative effect of niche width evolution on patterns of macroevolutionary rates.

Indeed the question of niche width evolution has been addressed in two important theory papers by Ackermann & Doebeli (2004 *Evolution*) and Ito & Shimada (2007 *Evol. Ecol. Res.*). These models look at phenotypic diversification and evolutionary branching, but did not include genetic mechanisms of reproductive isolation. Ackermann & Doebeli (2004) addressed the diversification of niche position (similar to one of our ecological traits) while allowing for directional selection on niche width. They showed that if larger niche width is beneficial, the evolution of niche width suppresses diversification (evolutionary branching) of niche position. If larger niche width is costly, branching in niche position occurs while niche width decreases along each branch. These findings are in line with the effect of increasing or decreasing competition width found in our model (cf. Figure 2b and d): if niche width increases, speciation would slow down during the second phase of diversification, with little effect on extinction rates, leading to lower long-term diversity; if niche width decreases, speciation would speed up during the second phase of diversification, with little effect on extinction rates, leading to higher long-term diversity.

Ito & Shimada (2007) investigated the coupled evolutionary branching of niche position and niche width. They found that branching of a species with narrow niche can occur on both niche position and niche width, with one of the branches first evolving large width, and subsequently narrow niche, hence two specialists. Thus, their results support a general pattern of narrower niche evolving across a range of phenotypes that diversify in their niche position. In our model, this would manifest as a gradual reduction of niche width (barring the episodic evolution of more generalist transient species). As before, decreasing niche width would accelerate speciation in the second phase of diversification, with little effect on extinction rates, and eventually result in higher long-term diversity.

In conclusion, if niche width evolves we expect the main alteration of our results to be the emergence of positive diversity dependence of speciation in the second phase of diversification (as more specialization evolves), and more shallow negative diversity dependence of speciation in the third phase (as predicted for small competition width). Given the relatively small effect of competition width on speciation that our model shows in diversification phase two, we expect that abiotic factors (the pace of landscape dynamics) would remain the dominant factor of speciation rates (and extinction rates) in this phase. A summary of this argument has been included in our revision (lines 346-359).

On a slightly different note, I believe that the authors can do more to be more specific about their conclusions. Currently the discussion presents a somewhat muddled message. One suggestion I'd make on this front would be, to use current phylogenetic methods on the simulated phylogenies, to test how different methods (e.g., w/ and w/o density dependent speciation and extinction rates) fare in a model where they know all the ingredients. That would put the discussion on a more solid footing (allowing the authors to make sharper statements), and serve as a validation of these methods (or not) in a model with known rates.

- We followed the Reviewer's suggestion and tested the capacity of available phylogenetic methods to detect the patterns of diversity dependence predicted by our model. We did this across the ranges of all parameters varied in the simulations. A general finding is a tendency for phylogenetic methods to over-detect negative diversity dependence in speciation rates. Spurious negative diversity-dependent speciation seems due to the confounding effect of the shift in speciation and extinction rates between diversification phases one and two. This is reported in new text in the Discussion (lines 432-448) and new figure S10 in the Supplementary Information.

The results thus call for the development of new phylogenetic models that distinguish between different temporal stages of diversity-dependence and diversity-independence. We emphasize that these models will allow to identify the different stages of diversification in a clade's history; estimate speciation and extinction rates across these stages; and test the model predictions regarding the ecological, genetic, and environmental controls of macroevolutionary rates in each stage (lines 462-467).

Related to the that, a clarification question: it wasn't obvious to me, in the description of species delimitation in the model, what the time-steps were. Are the authors sampling the populations in each time-step (after each reproduction event) or do they coarse-grain in time? Seems like that might affect the phylogenetic relationships being inferred.

- We extracted the composition of the metacommunity every 100 generations, and used these data to reconstruct the phylogenetic relationships among species. In a previous study (Gascuel *et al.* 2015 *Systematics Biology*) we showed that similar relationships between species were obtained with a sampling interval of 10 generations. Thus, our conclusions do not depend on the time interval used, provided that it is short enough. We pointed this out in the revision (lines 827-832).

Overall, I believe this is a potentially positive contribution, but would benefit from a slightly less grandiose framing as well as some robustness checks of the simulation model.

- We thank the Reviewer for her/his helpful advice. The manuscript has been largely rewritten to focus on the question of biotic vs. abiotic drivers of macroevolutionary rates, rather than Darwin's principle of divergence. In the Discussion, we addressed the effect of resource overlap (in the light of our and other previous studies, our results likely unaffected as long as resource overlap is neither too small nor too large) and the potential effect of niche width evolution (which we predict to cause positive diversity dependent speciation in phase two, but should not change the conclusion that the pace of landscape dynamics is the dominant factor of both speciation and extinction rates past the initial radiation). We also performed the phylogenetic tests suggested by the Reviewer; the results highlight the limitation of current methods at correctly detecting patterns of diversity dependence, and call for the development of new phylogenetic models to reveal the patterns of variation in macroevolutionary rates that our model predicts.

Reviewers' comments:

Reviewer #1

Please note that while Reviewer 1 doesn't have remarks to the author, in his/her remarks to the editor, Reviewer 1 said he/she is happy with the revision.

Reviewer #3 (Remarks to the Author):

My impression of the first version of this manuscript was that it was interesting, and a potentially good contribution to speciation literature, but suffered from a few deficiencies, mainly (i) the framing around Darwin's principle of divergence seemed to be completely off, ignoring decades of theory on competition and macro-evolutionary patterns, and (ii) some model assumptions and parameters seemed crucial for the results and it seemed warranted to check the robustness of the results to perturbations of these parameters. In this revision, the authors partially address my concerns. I think the paper has improved, though can be improved further still.

On (i), the authors revised the title, abstract, and the introduction of the paper to motivate the study in less historically and more using modern concepts. This framing I think is more accurate and helps to better put the paper in the current context of speciation research. I think this improves the paper (although the claim that Darwin's principle of divergence was dead until this paper rescued it still is buried in the discussion), and I am satisfied with the changes here.

I am a bit less satisfied on (ii), where the authors acknowledge the concerns I raised (arrangement of the sites and resources and the overlap between the resource distribution of different sites), but chose to refer to previous literature. In particular the authors acknowledge (for example) that too little resource overlap will lead to lineages having a harder time to colonize new sites. That is clearly true (and I made the same point in my review), but it seems to me that this fact may well have quantitative consequences for the dynamics of diversity over time, e.g., it would likely mean that most species arise locally. To me, this still does seem like a relevant robustness check, given that the intuitively it would interact with the other main factors studies, such as the frequency of isolation, and possibly with the underlying genetics.

The authors interpret my comment about the regular distribution as applying mainly to the number of sites, but I really had meant the regular distribution of sites along a resource gradient (so that the sites that are next to each other have the most similar resources, and sites farther away have more different). That's not necessarily is going to be the case in nature at the macro-evolutionary relevant spatial scales, and given that this again can affect dynamics of diversity (and quantitative predictions on these features is the main selling point of the paper) I feel this is not to be easily waved away.

That said, I think the addition of a discussion of these issues into the Discussion section is an improvement. I am also pleased that the authors took up my suggestion to use existing phylogenetic methods on simulated species phylogenies. I think the result that these methods tend to overestimate the frequency of negative density dependent speciation is interesting, though I am less sanguine than the authors about the resolution of the problem.

Manuscript #17-21576 entitled “How ecological competition, genetic divergence, and landscape dynamics control the macroevolutionary rates of speciation and extinction”, by R. Aguilée, F. Gascuel, A. Lambert and R. Ferriere

Response to Reviewers

- There were no remarks from Reviewers #1 and #2.

Response to Reviewer #3 (Remarks to the Author):

My impression of the first version of this manuscript was that it was interesting, and a potentially good contribution to speciation literature, but suffered from a few deficiencies, mainly (i) the framing around Darwin's principle of divergence seemed to be completely off, ignoring decades of theory on competition and macro-evolutionary patterns, and (ii) some model assumptions and parameters seemed crucial for the results and it seemed warranted to check the robustness of the results to perturbations of these parameters. In this revision, the authors partially address my concerns. I think the paper has improved, though can be improved further still.

On (i), the authors revised the title, abstract, and the introduction of the paper to motivate the study in less historically and more using modern concepts. This framing I think is more accurate and helps to better put the paper in the current context of speciation research. I think this improves the paper (although the claim that Darwin's principle of divergence was dead until this paper rescued it still is buried in the discussion), and I am satisfied with the changes here.

- We acknowledge the Reviewer's appreciation of our revision and thank her/him again for giving us the impetus for such an alternative and improved presentation.

I am a bit less satisfied on (ii), where the authors acknowledge the concerns I raised (arrangement of the sites and resources and the overlap between the resource distribution of different sites), but chose to refer to previous literature. In particular the authors acknowledge (for example) that too little resource overlap will lead to lineages having a harder time to colonize new sites. That is clearly true (and I made the same point in my review), but it seems to me that this fact may well have quantitative consequences for the dynamics of diversity over time, e.g., it would likely mean that most species arise locally. To me, this still does seem like a relevant robustness check, given that the intuitively it would interact with the other main factors studied, such as the frequency of isolation, and possibly with the underlying genetics.

The authors interpret my comment about the regular distribution as applying mainly to the number of sites, but I really had meant the regular distribution of sites along a resource gradient (so that the sites that are next to each other have the most similar resources, and sites farther away have more different). That's not necessarily going to be the case in nature at the macro-evolutionary relevant spatial scales, and given that this again can affect dynamics of diversity (and quantitative predictions on these features is the main selling point of the paper) I feel this is not to be easily waved away.

- We have now performed the robustness checks requested by the Reviewer. We relaxed the assumption of a correlation between the distribution of ecological optima and the geographical location of sites, and we explored the effect of the degree of overlap between resource distributions in adjacent sites by varying the phenotypic distance between ecological optima. To do so we generated 84 landscapes for which the ecological optima of the 9 sites were randomly

chosen in uniform distributions of various widths. The phenotypic distance between the optima of adjacent sites (and thus the degree of resource distributions overlap of adjacent sites which varies in the opposite direction) is different for each pair of sites, and there is no correlation between geographical proximity and ecological similarity. The figure below (new Supplementary Figure 1) shows the reference landscape and two of these random landscapes as examples.

Left column: Reference landscape (phenotypic distance between ecological optima of adjacent sites: 1 unit). *Middle and right columns:* Two random landscapes with different mean phenotypic distance between ecological optima of adjacent sites (middle column: 0.69 unit; right column: 3.31 units). *Top row:* Relation between phenotypic proximity of sites (numbered from 1 to 9) and geographical location (geographically adjacent sites are linked by an edge). *Bottom row:* Global distribution of resources if all sites were merged together.

The results of this new set of simulations show that:

- 1) The diversification mechanism described in the manuscript operates even when geographical position and ecological similarity of sites are uncorrelated.
- 2) The diversification mechanism operates when the overlap of resource distributions between adjacent sites is neither too small (in that case, colonization of adjacent sites fails) nor too large (in that case, species in adjacent sites are not sufficiently differentiated at secondary contact to coexist on the long term).
- 3) For the range of overlap of resource distributions under which the diversification mechanism operates, the larger the distance between ecological optima (i.e. the smaller the overlap of resource distributions) the higher the diversification rate in stages 1 and 2 (because allopatric divergence is then more efficient) and the lower the stationary turnover (because speciation is

less likely, due to failed colonization of adjacent sites; and extinction is less frequent, because species at secondary contact are more differentiated).

These new results both refine and extend our previous conclusions. In our revision, we introduce the randomized landscape version of the model in the Introduction (ll. 89-92), Methods (ll. 706-710), and new Supplementary Figure 1. We incorporated the new results in the Discussion (ll. 343-354) and presented the corresponding simulations in the new Supplementary Figure 11.

That said, I think the addition of a discussion of these issues into the Discussion section is an improvement. I am also pleased that the authors took up my suggestion to use existing phylogenetic methods on simulated species phylogenies. I think the result that these methods tend to overestimate the frequency of negative density dependent speciation is interesting, though I am less sanguine than the authors about the resolution of the problem.

- Again, we thank the Reviewer for the time s/he has invested in evaluating our manuscript and their insightful comments and very helpful suggestions to improve our work.

REVIEWERS' COMMENTS:

Reviewer #3 (Remarks to the Author):

In the previous version, I found the paper much improved but had suggested some of the robustness checks that I recommended in the first round really needed to be done. In this version, the authors have undertaken these robustness checks, in particular, by generating non-regular landscapes with different amounts of resource overlap and correlation between geographical and resource distances. The results of these checks, as the authors say, both support and qualify the results from the regular landscapes. I especially like Figure S11 that reports the results from landscape where resource optima in each site is randomly (independent of its location) chosen. The hump-shaped patterns in diversification rates in stages 1 and 2 (and the uniform decrease in turnover at stationarity) are to me quite interesting patterns. I might even suggest moving that figure to the main text.

Regardless, I think at this point, the paper is much improved and a good contribution to the literature. I enjoyed reading it, and I think many others will, too.

Manuscript #17-21576B entitled “Clade diversification dynamics and the biotic and abiotic controls of speciation and extinction rates”, by R. Aguilée, F. Gascuel, A. Lambert and R. Ferriere

Response to Reviewers

- There were no remarks from Reviewers #1 and #2.

Response to Reviewer #3 (Remarks to the Author):

In the previous version, I found the paper much improved but had suggested some of the robustness checks that I recommended in the first round really needed to be done. In this version, the authors have undertaken these robustness checks, in particular, by generating non-regular landscapes with different amounts of resource overlap and correlation between geographical and resource distances. The results of these checks, as the authors say, both support and qualify the results from the regular landscapes. I especially like Figure S11 that reports the results from landscape where resource optima in each site is randomly (independent of its location) chosen. The hump-shaped patterns in diversification rates in stages 1 and 2 (and the uniform decrease in turnover at stationarity) are to me quite interesting patterns. I might even suggest moving that figure to the main text.

Regardless, I think at this point, the paper is much improved and a good contribution to the literature. I enjoyed reading it, and I think many others will, too.

- We thank the Reviewer for the time s/he has invested in evaluating our manuscript and their insightful comments and very helpful suggestions to improve our work.
- As suggested, we moved the Supplementary Figure 11 to the main text (now Figure 5). The text was left unchanged.